# **kgbench**: A Collection of Knowledge Graph Datasets for Evaluating Relational and Multimodal Machine Learning

Peter Bloem⋆[0000−0002−0189−5817], Xander Wilcke⋆[0000−0003−2415−8438],
Lucas van Berkel⋆[0000−0002−2524−1279], and Victor de Boer[0000−0001−9079−039X]

Informatics institute, Vrije Universiteit Amsterdam, the Netherlands
{p.bloem, w.x.wilcke, v.de.boer}@vu.nl, l.l2.van.berkel@student.vu.nl

**Abstract.** Graph neural networks and other machine learning models offer a promising direction for interpretable machine learning on relational and multimodal data. Until now, however, progress in this area is difficult to gauge. This is primarily due to a limited number of datasets with (a) a high enough number of labeled nodes in the test set for precise measurement of performance, and (b) a rich enough variety of multimodal information to learn from. We introduce a set of new benchmark tasks for node classification on RDF-encoded knowledge graphs. We focus primarily on node classification, since this setting cannot be solved purely by node embedding models. For each dataset, we provide test and validation sets of at least 1 000 instances, with some over 10 000. Each task can be performed in a purely relational manner, or with multimodal information. All datasets are packaged in a CSV format that is easily consumable in any machine learning environment, together with the original source data in RDF and pre-processing code for full provenance. We provide code for loading the data into `numpy` and `pytorch`. We compute performance for several baseline models.

**Keywords:** knowledge graphs · machine learning · message passing models · multimodal learning

## 1 Introduction

The combination of knowledge graphs and machine learning is a promising direction. In particular, the class of machine learning models known as *message passing models* offer an interesting set of abilities [1,35]. These models operate by propagating information along the structure of the graph and are trained end-to-end, meaning all information in the data can potentially be used if it benefits the task. Even the contents of the literals may be used by attaching encoder networks to learn how literals should be read, leading to an end-to-end model for multimodal learning on knowledge graphs. The message passing framework is also a promising direction for interpretable machine learning, as the computation of the model can be directly related to the relational structure of the data [9].

---

⋆ The first three authors contributed equally to this paper.

Unfortunately, the progress of message passing models and related machine learning approaches has been difficult to gauge, due to the lack of high quality datasets. Machine learning on knowledge graphs is commonly evaluated with two abstract tasks: *link prediction* and *node labeling*. In the latter, the model is given the whole graph during training, together with labels for a subset of its nodes. The task is to label a set of withheld nodes with a target label: a class for node classification or a number for node regression.

While link prediction is probably more popular in recent literature, node labeling is more promising for developing message passing models. In link prediction, it is not clear whether message passing models offer an advantage over embedding models on currently popular benchmarks, without a considerable increase in computational requirements. In node labeling, however, the task cannot be solved from node embeddings alone. In some way, the deeper structure of the graph *needs* to be taken into account, making it a better testing ground for message-passing algorithms such as R-GCNs [28] and R-GATs [6].

In this work, we specifically focus on knowledge graphs that are built on top of the *Resource Description Framework* (RDF). The most common datasets used in node classification on such knowledge graphs, are the AIFB, MUTAG, BGS and AM datasets, which were first collected and published for this purpose in [22]. Their details are given in Table 1. These datasets are well suited to message passing methods since they are relatively small, allowing a message passing model to be trained full-batch so that we can gauge the performance of the model independent the influence of minibatching schemes. However, this small size of the graphs also means a small number of labeled instances, and, in particular, a small *test set*, sometimes with less than 50 instances.

While limited training data is often a cause for concern in machine learning, limited test data is usually the greater evil. With limited training data, we may have a model that fails to perform well, but with limited test data we cannot even tell how well our model is performing. In statistical terms: a performance metric like accuracy is an estimate of a true value, the expected accuracy under the data distribution, based on a sample from that distribution; the test set. The larger that sample, the more accurate our estimate, and the smaller our uncertainty about that estimate. Figure 1 shows the size of the 95% confidence intervals for different test set sizes on a balanced binary classification problem. We see that only at 10 000 instances do we have sufficient certainty to say that a model with a measured accuracy of 0.94 is most likely better than one with a

**Table 1.** The currently most commonly used benchmark datasets for node classification.

| Dataset | AIFB | MUTAG | BGS* | AM* |
|---|---|---|---|---|
| Entities | 8 285 | 23 644 | 87 688 | 246 728 |
| Relations | 45 | 23 | 70 | 122 |
| Edges | 29 043 | 74 227 | 230 698 | 875 946 |
| Labeled | 176 | 340 | 146 | 1 000 |
| Classes | 4 | 2 | 2 | 11 |

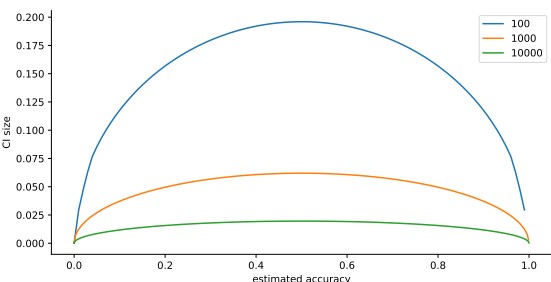

**Fig. 1.** The size of a 95% confidence interval around an estimate of accuracy for a two-class problem, with test sets of 100, 1 000 and 10 000 instances. Note that 10 000 instances are required before we can tell apart all estimates that differ by 0.01.

measured accuracy of 0.93. The test set sizes in Table 1 do not allow for anything but the most rudimentary discrimination.

Additionally, while these datasets provide some multimodal data in the form of literals, they are usually not annotated with datatypes, the modalities remain restricted to simple strings containing natural language, or structured information like numerical values and dates. Richer multimodal information like images, audio or even video would present a more exciting challenge for the possibility of integrating such data in a single end-to-end machine learning model.

To overcome these problems, we introduce kgbench: a collection of evaluation datasets knowledge graph node labeling. Each comes with a test set of between 2 000 and 20 000 labeled nodes, allowing for precise estimates of performance.

Each dataset can be used in two different ways. In the **relational** setting, each node is treated as an atomic object, with literals considered equal if their lexical content is equal. This mode can be used to evaluate relational machine learning models, as in [28,6,23]. In the **multimodal** setting, the *content* of literal nodes is taken into account as well, as described in [34,35]. In addition, each dataset can also be used to evaluate link prediction models by ignoring the node labels (see Section 2.3 for details).

The datasets are offered as RDF, with each dataset packaged both in N-Triples and in HDT [10] format. Additionally, since loading RDF into machine learning environments can be non-trivial, we offer pre-processed versions of each dataset, which contain integer indices for all nodes and relations in the graph. These are stored as a set of CSV files, to ensure that they can be directly read by a large number of machine learning libraries. We also provide explicit dataloading code for Numpy and Pytorch, as well as scripts to converts any RDF-encoded knowledge graph to this format.

All data and code is hosted on Github.[1] To ensure long term availability and to provide a permanent identifier, snapshots are also hosted on Zenodo.[2] Each

---

[1] https://github.com/pbloem/kgbench

dataset is licensed under the most permissive conditions allowed by the licenses on the source datasets.

## 1.1   Related work

Similar efforts to ours include: CoDEx [25], a link prediction benchmark, including multilingual literals and RichPedia, [32] a large-scale multimodal knowledge graph, with no specific machine learning task attached. Other link prediction research has included new benchmark data [30,15]. Our datasets are, to the best of our knowledge the first node labelling benchmarks that focus on large test set size and multimodal learning. In [22] node labeling tasks on large knowledge graphs are included, but the number of total instances in the dataset never exceeds $2\,000$, and canonical snapshots of the knowledge graphs are not provided.

The field of knowledge graph modeling by machine learning methods can be divided into two main camps: pure embedding methods, which learn node embeddings directly, and message passing approaches which learn from the graph structure more explicitly. For pure embedding methods [24] serves as a good overview of the state of the art. Message passing methods are popular [1], but in the specific domain of knowledge graphs, there has been less progress, with R-GCNs [28] and R-GATs [6] as the main approaches. Other approaches include kernel methods [7] and feature-extraction approaches [23].

## 2   Method

In this section we detail the main design choices made in constructing the tasks and datasets in kgbench. Our data model in all cases follows RDF. That is, a knowledge graph is defined as a tuple $G = (V, R, E)$, with a finite set of nodes $V$, a finite set of relations $R$ and a finite set of edges (also known as *triples*) $E \subseteq V \times R \times V$. The nodes in $V$ can be atomic entities,[3] or *literals*, defined by a string which is optionally tagged with a *string annotation*. This annotation can either be a *datatype* (an IRI expressing the type of data) or a *language tag*.

### 2.1   Desiderata

A good machine learning benchmark must satisfy a large number of constraints. We have focused primarily on the following.

**Large test sets** A large test set is essential for accurate performance estimates. This is our primary concern.

**Manageable graph size** A small benchmark dataset allows for quick evaluation of hypotheses and quick iteration of model designs, and keeps machine learning research accessible.

---

[2] Details, including DOIs, under the following references amplus [3], dmgfull and dmg777k [36], dblp [4], mdgenre and mdgender [2].

[3] Entities may be *resources*, identified by an IRI, or *blank nodes*.

**Small training sets** Keeping the number of training instances relatively low has several benefits: it leaves more instances for the validation and test sets and it makes the task more difficult. If the instances are more sparsely labeled, models are forced to use the graph structure to generalize. It is common practice, once hyperparameter tuning is finished, to combine the training and validation sets into a larger training set for the final run. Normally this conveys only a very small extra advantage. In our case, adding the validation data often has a very large effect on how easy the task becomes, and which structure can be used to solve. **For this reason, in our tasks, practitioners should only ever train on the training data, no matter what set is being evaluated on.**

**Multimodal literals** Where possible we offer literals of multiple modalities. We annotate existing strings with datatypes and language tags, and add images and spatial geometries. These are placed into the graph as literals rather than as hyperlinks, making the dataset self-contained.

## 2.2   Data Splitting and Layout

Each dataset provides a canonical training/validation/test split. We also split off a meta-test set if the data allows. This is an additional set of withheld data. It serves as an additional test set for review studies over multiple already-published models. This provides the possibility to test for overfitting on the test set if the dataset becomes popular. **Any practitioner introducing a single new model or approach, should ignore the meta-test set.** [4]

Each dataset is provided as an RDF graph, with the target labels kept in separate files. We emphatically choose *not* to include the target labels in the dataset, as this would then require practitioners to manually remove them prior to training, which creates a considerable risk of data leakage.

*Preprocessing* The most common preprocessing step for relational machine learning is to map all relations and nodes to integer indices. We have preprocessed all datasets in this manner and provided them as a set of CSV files (in addition to the original RDF). While a collection of CSV files may not be in keeping with the spirit of the Semantic Web, this format greatly facilitates reading the data into any any data science or machine learning software, without the need to parse RDF or load the data into a triple store.

This format also allows practitoners to choose between the relational and multimodal setting in a simple manner. If only the integer indices are read, then the data is viewed purely from a relational setting. The mappings from the integer indices to the string representations of the nodes then provide the multimodal layer on top of the relational setting.

---

[4] It is common practice to not publish the meta-test set to ensure that it is not used by practitioners until it is necessary. In our case this makes little sense, since the meta-test set could easily be derived from the available raw data manually.

```
1  @prefix :     <http://kgbench.info/dt#> .
2  @prefix rdfs: <http://www.w3.org/2000/01/rdf-schema#> .
3  @prefix xsd:  <http://www.w3.org/2001/XMLSchema#> .
4
5  :base64Image a rdfs:Datatype ;
6      rdfs:subClassOf xsd:base64Binary ;
7      rdfs:label "Base64-encoded image"@en ;
8      rdfs:comment "An image encoded as a base64 string"@en .
9
10 :base64Video a rdfs:Datatype ;
11     rdfs:subClassOf xsd:base64Binary ;
12     rdfs:label "Base64-encoded video"@en ;
13     rdfs:comment "A video encoded as a base64 string"@en .
14
15 :base64Audio a rdfs:Datatype ;
16     rdfs:subClassOf xsd:base64Binary ;
17     rdfs:label "Base64-encoded audio"@en ;
18     rdfs:comment "An audio sequence encoded as a base64 string"@en .
```

**Listing 1.1.** A small ontology (kgbench.info/dt.ttl) for base64-encoded image, audio, and video.

### 2.3   Link prediction

Our focus is node labeling, but since link prediction is an unsupervised task, each of our datasets can also be used in link prediction, both for purely relational settings and for multimodal settings. In such cases, we suggest that the following guidelines should be followed:

- The triples should be shuffled before splitting. The validation, test and meta-test set should each contain 20 000 triples, with the remainder used for training. We include such a split for every dataset.
- In contrast to the node labeling setting, we do not enforce limited training data. The final training may be performed on the combined training and validation sets, and tested on the test set.
- Practitioners should state that the data is being *adapted* for link prediction, and whether the dataset is being used in relational or in multimodal setting.

### 2.4   Expressing Binary Large Objects

No convention currently exists for encoding images, videos, or audio in literals. A convention in the realm of relational databases is to store complex datatypes as Binary Large OBjects (BLOBs). Here, we chose to adopt this convention by encoding binary data in base64 encoded string literals. The conversion to and from binary data is well supported by many popular programming languages.

To express that a certain string literal encodes a complex type it should be annotated as such using a suitable datatype. The straightforward choice for this datatype would be `xsd:base64Binary`. However, this does little to convey the type of information which it encodes, which makes it difficult to build machine learning models that distinguish between these types. To accommodate this

**Table 2.** Statistics for all datasets. We consider a dataset "GPU friendly" if the R-GCN baseline can be trained on it with under 12 GB of memory and "CPU-friendly" if this can be done with under 64 GB. mdgender is not meant for evaluation (see Section 5.1).

| Dataset | | amplus | dmgfull | dmg777k | dblp | mdgenre | mdgender* |
|---|---|---|---|---|---|---|---|
| Triples | | 2 521 046 | 1 850 451 | 777 124 | 21 985 048 | 1 252 247 | 1 203 789 |
| Relations | | 33 | 62 | 60 | 68 | 154 | 154 |
| Nodes. | | 1 153 679 | 842 550 | 341 270 | 4 470 778 | 349 344 | 349 347 |
| | entities | 1 026 162 | 262 494 | 148 127 | 4 231 513 | 191 135 | 191 138 |
| | literals | 127 517 | 580 056 | 192 143 | 239 265 | 158 209 | 158 209 |
| Density | | $2 \cdot 10^{-6}$ | $3 \cdot 10^{-6}$ | $7 \cdot 10^{-6}$ | $1 \cdot 10^{-6}$ | $1 \cdot 10^{-5}$ | $1 \cdot 10^{-5}$ |
| Degree | avg | 4.37 | 4.47 | 4.53 | 9.83 | 7.17 | 6.89 |
| | min | 1 | 1 | 1 | 1 | 1 | 1 |
| | max | 154 828 | 121 217 | 65 576 | 3 364 084 | 57 363 | 57 363 |
| Classes | | 8 | 14 | 5 | 2 | 12 | 9 |
| Labeled | total | 73 423 | 63 565 | 8 399 | 86 535 | 8 863 | 57 323 |
| | train | 13 423 | 23 566 | 5 394 | 26 535 | 3 846 | 27 308 |
| | valid | 20 000 | 10 001 | 1 001 | 20 000 | 1 006 | 10 005 |
| | test | 20 000 | 20 001 | 2 001 | 20 000 | 3 005 | 10 003 |
| | meta | 20 000 | 10 001 | | 20 000 | 1 006 | 10 007 |
| Source | | [5] | see text | | [29][31][21] | [31][12] | [31][12] |
| GPU friendly | | ✓ | | ✓ | | | |
| CPU friendly | | ✓ | ✓ | ✓ | | ✓ | ✓ |
| Datatypes[6] | | | | | | | |
| Numerical | | 8 418 | 64 184 | 8 891 | | 1 387 | 1 387 |
| Temporal | | 6 676. | 463 | 290 | | 37 442 | 37 442 |
| Textual | | 56 202 | 340 396 | 117 062 | 239 265 | 51 852 | 51 852 |
| Visual | | 56 130 | 58 791 | 46 061 | | 67 528 | 67 528 |
| Spatial | | | 116 220 | 20 837 | | | |

distinction, we instead introduce a small collection of datatype classes to annotate binary-encoded strings in accordance with their information type (Listing 1.1).[5]

## 3  Datasets

Table 2 lists the datasets contained in kgbench and their basic statistics, as well as an overview of the distribution of modalities per dataset. All datasets were created by combining publicly available data sources, with no manual annotation. Enrichment was limited to combining data sources, and annotating literals.

### 3.1  The Amsterdam Museum Dataset (amplus)

The Amsterdam Museum is dedicated to the history of Amsterdam. Its catalog has been translated to linked open data [5]. The AM dataset, as described in

---

[5] The same may be achieved with additional triples. While this would remove the need for new datatypes, it would render the isolated literal meaningless. This contrasts with most other datatypes, which still convey their meaning in isolation.

[6] Numerical includes all subsets of real numbers, as well as booleans, whereas date, years, and other similar types are listed under temporal information. Textual includes the set of strings (possibly without datatype, its subsets, and raw URIs (e.g. links). Images and geometries are listed under visual and spatial information, respectively.

**Table 3.** The class mapping for the `amplus` data. The original categories are translated from their original Dutch names.

| original | class | frequency |
|---|---|---|
| Furniture, Glass, Textile, Ceramics, Sculpture, Arts & crafts | Decorative art | 25 782 |
| Prints | Prints | 22 048 |
| Coins & tokens, Archaeological artifacts, Measures & weights | Historical artifacts | 7 558 |
| Drawings | Drawings | 5 455 |
| Non-noble metals art, Noble metal art | Metallic art | 4 333 |
| Books, Documents | Books & documents | 4 012 |
| Paintings | Paintings | 2 672 |
| Photographs | Photographs | 1 563 |

Table 1 is already established as a benchmark for node classification: the task is to predict the type of a given collection item.

In this version, the number of labeled instances is arbitrarily limited to 1000, resulting in small test set sizes. We return to the original data and make the following changes: we collect all collection items as instances, annotate a large number of literals with the correct datatype, and insert images as base64 encoded literals. We also include only a subset of the relations of the original data: to make the dataset both small and challenging. Finally, we remap the categories to a smaller set of classes; to create a more balanced class distribution. The mapping is given in Table 3.

The `amplus` data is provided under a Creative Commons CC-BY license.

### 3.2   The Dutch Monument Graph (`dmgfull`, `dmg777k`)

Like `amplus`, the Dutch Monument Graph (DMG) is a dataset from the Digital Humanities. Encompassing knowledge from several organizations, the DMG contains information about 63 566 registered monuments in the Netherlands.

Engineered with the goal of creating a highly multimodal dataset, the DMG contains information in six modalities, five of which are encoded as literals. This includes the often common numerical, temporal, and textual information, but also visual information in the form of images, and, more uniquely, several different kinds of spatial information. Taken all together, these modalities provide the monuments with a diverse multimodal context which includes, amongst other things, a short title, a longer description, a construction date, the city and municipality it lies in, several images from different directions, a set of geo-referenced coordinates, and a polygon describing its footprint.

Unique of this dataset is the strong presence of geospatial information. This form of information encompasses the spatial and hierarchical relations between spatial features, such as cities, municipalities, and countries, as well as sets of coordinates in $\mathbb{R}^d$ which describe a position and/or shape. These coordinates are expressed using the *well-known text* format (WKT), and are linked using the *Open GeoSpatial Consortium*'s GeoSPARQL ontology.

Five different knowledge graphs from four different organizations,[7] were combined to form the DMG. The information from these organizations was combined using entity resolution based on string comparison, matching municipality and city names, as well as multi-part addresses. Once merged, the information was cleaned and provided with accurate class and datatype declarations where missing.

The *777k*-variant is a subset encompassing $8\,399$ monuments created by sampling monuments from the top-5 monument classes that have no missing values. Both datasets are published under the CC-BY license.

### 3.3    The Movie Dataset (`mdgenre`, `mdgender`)

The Movie datasets are subsets of Wikidata [31] in the movie domain. We select any movies that are recorded as ever having won or been nominated for an award. Every person affiliated with any of these movies is also selected if the relation between the movie and the person is in a whitelist.

This whitelist consists of relations that satisfy the following conditions: every relation needs to have a Wikidata prefix and the relations do not direct to an identifier (ID) tag outside of Wikidata. Every triple that contains a movie or individual on their respective lists and a relation on the whitelist is extracted. This creates a knowledge graph that is centred around movie-related data and has a longest path of 4, making the knowledge graph relatively simple.

The main objective of this dataset is to predict the genre of the movies. Movies can have multiple genres, which is not practical when creating a single-label classification problem. Therefore, movies are assigned a genre based on a solution to the Set Cover Problem, which was derived using [38]. Each movie is assigned a single genre of which it already was part. This simplifies the multi-label classification objective to a multiclass classification objective. Additionally, the Movie Datasets also contain a gender objective, which we include as a sanity check as the objective is considered easier compared to the genre objective (see Section 5.1 for a discussion). As the classification in the Wikidata knowledge base is already suitable for multiclass classification, no further constraining as with the genres was not necessary.

We download thumbnail images from URLs in Wikidata and include base64-encoded representations. We also include thumbnails of images in the Internet Movie Database: using the IMDb-identifier in Wikidata, the respective web page at the IMDb-website is obtained for their thumbnail, which is in turn downloaded, converted and inserted in the same way.

The relational data in these datasets is taken from Wikidata, and provided under the same CC0/Public domain license that applies to Wikidata. For $40\,449$ out of the $68\,247$ images in this dataset, we extracted thumbnails from larger images published by the Internet Movie Database. The copyright of the original images resides with their producers. We assert no rights on this part of the data

---

[7] (1) the Dutch Cultural Heritage Agency, www.cultureelerfgoed.nl, (2) the Dutch Cadastre, Land Registry and Mapping Agency, www.kadaster.nl, (3) Statistics Netherlands, www.cbs.nl, and (4) Geonames, www.geonames.org.

for redistribution or use outside non-commercial research settings. The remainder of the thumbnails is taken from from the Wikimedia repository, and distributed under the individual license of each image.

### 3.4   The DBLP Dataset (`dblp`)

The DBLP repository [29] is a large bibliographic database of publications in the domain of computer science. This was converted to RDF under the name L3S DBLP, of which we used the HDT dump[8]. To provide a classification task on this data we extracted citation counts from the OpenCitations project [21], using the REST API. We checked all DOIs of papers in the DBLP dump, giving us a set of 86 535 DOIs that are present in both databases. These are our instances.

   We also extract information from Wikidata about researchers. We use the XML dump of DBLP [29] to extract ORCiDs, which allows us to link 62 774 people to Wikidata. For each person linked, we extract triples from the one-hop neighborhood in Wikidata. We use 24 relations from the DBLP data and 44 relations from Wikidata.

   Since we are focusing here on classification tasks, we turn the prediction of the citation count into two classes: those papers which received one citation, and those wich received more (due to the skewed distribution this the closest to a median-split). We have also preserved the original citation counts in the data, so the task can also be treated as a node regression task. This dataset is provided under a CC0/Public domain license.

## 4   Code and Baselines

In addition to the datasets in their RDF and CSV formats, we also provide scripts to convert any arbitrary RDF-encoded graph to our CSV format. To import these datasets into a machine learning workflow, we further provide a small Python library that loads any dataset that makes use of our CSV format into a object containing Pytorch [20] or Numpy [19] tensors, together with mappings to the string representations of the nodes. This provides both a utility sufficient for the majority of current machine learning practice, and a reference implementation for any setting where such a dataloader does not suffice.

   In addition to the new datasets of Table 2, the repository also includes legacy datasets `aifb` and the original Amsterdam Museum data, named `am1k` here. These are useful for debugging purposes.

   The dataloader allows the data to be loaded in a single function call. It also provides utility functions for pruning the dataset to a fixed distance around the instance nodes, and for re-ordering the nodes so that the datatypes are ordered together (which may reduce expensive tensor indexing operation in implementing multimodal models). We also provide three baseline models as reference for how to use the data in practice:

---

[8] Available at https://www.rdfhdt.org/datasets/.

**Table 4.** Performance of baselines on the datasets in the collection. The RGCNs could not be trained on `dblp` in under 64Gb of memory.

| setting | baseline | amplus | dmgfull | dmg777k | dblp | mdgenre |
|---------|----------|--------|---------|---------|------|---------|
| relational | Features | 0.72 | 0.73 | 0.42 | 0.72 | 0.66 |
|  | R-GCN | 0.77 | 0.71 | 0.70 | - | 0.63 |
| multimodal | MR-GCN | 0.86 | 0.76 | 0.57 | - | 0.62 |

**Features** This model extracts binary graph features about the set of triples incident to the instance node, which are then used by a logistic regression classifier. Over the whole set of training instances, all of the following binary features are considered: (a) whether a particular predicate $p$ is present or not, (b) whether a particular predicate is present in a specific direction, i.e. outgoing or incoming, and (c) whether a particular predicate, in a particular direction, connects the instance node to a specific node $n$. For all collected features, the information gain is computed for splitting the training instances on that feature. The $k$ features with the highest information gain are kept and used to train a classifier.

**R-GCN** The default classification R-GCN model [28]. It contains two R-GCN layers that are fed with a one-hot encoding of the nodes, mapping to a hidden layer, which is mapped to class probabilities. By default, a hidden size of 16 is used, with a basis decomposition of 40 bases. This baseline is purely relational, and ignores multimodal information.

**MR-GCN** We provide a stripped-down version of the MR-GCN model [34]. Unlike the original, this model does not train its feature extractors end-to-end, which means that no backpropagation is needed beyond the R-GCN layer, saving memory. The literal features are extracted by pretrained models: a Mobilenet-v2 [26] for the images and DistilBERT [27] for literals. After feature extraction, the features are scaled down to a uniform input dimension $d$ by principal component analysis.

### 4.1   Baseline Performance

Table 4 shows the accuracies of the three baseline models on the datasets in `kgbench`. The R-GCN models were trained for 50 epochs with default hyperparameters. That is, a two-layer model, with ReLU activation and a hidden size of 16. Training was done full-batch for 50 epochs with the Adam optimizer with default parameters and a learning rate of 0.01. A $0.5 \cdot 10^{-3}$ L2 penalty was applied to the weights of the first layer. The features baseline was run with $k = 2\,000$ and a logistic regression classifier with no regularization.

These numbers should be taken as broad baselines for how default models perform on these datasets, and not as the last word of the performance of, for instance, the R-GCN. It may well be possible to achieve better performance with more extensive hyperparameter tuning, a different architecture, or more training epochs. In particular, the MR-GCN used here is likely considerably less performant than the fully end-to-end version.

## 5   Discussion

### 5.1   Broader impact

While only a small proportion of benchmarks that are published achieve broad community-wide uptake, those that do ultimately have a profound impact on the direction in which technology is developed. A dataset like ImageNet [8] was developed in a time when no models were available that could solve the task, but it is now commonly used to pretrain computer vision models that are widely distributed and used in production systems. Even a dataset like FFHQ [13], which was specifically compiled with diversity and representation in mind has led to pre-trained models that contain bias, which is ultimately exposed in downstream applications [17].

For this reason we consider it wise to discuss both the biases present in the data and the implications of setting certain labels as training targets.

**Bias in training data**  A common source of discussion in AI Ethics is the bias present in training data, especially where the representation of people is concerned [16]. A case in point are the `mdgenre` and `dblp` datasets, which both contain the "Sex or Gender" property of Wikidata.[9] In the former, a disproportionate number of the actors in the data are men. While this may be an accurate reflection of a bias in the world,[10] it means that actions taken based on the predictions of a production model trained on this data, may end up amplifying the data biases.

We have chosen not to de-bias the data for various reasons. First, we can only correct for the biases for which we have attributes (such as sex, gender, race, or religion). Second, even if we resample in this way, the biases may still manifest, for instance in the completeness of the data for men and women. Finally, debiasing the data ourselves, by a fixed strategy removes the possbility of investigating the debiasing method itself.

In short, we take it as a given that the data is biased. Since the data was largely retrieved completely as found in the wild, with only crude filtering based on node neighborhoods and relation whitelists, we may assume that these biases are reflective of the biases in real-world data. This may be used to study data bias in knowledge graphs, but any model trained on these datasets should not be put into production without careful consideration.

**Choice of target relations**  In all cases, our primary reasons for setting a particular target relation are technical. It is challenging to find a set of classes that are well-balanced, offer a large amount of instances, and provide a challenging task. Moreover, in the multimodal setting, a variety of literals with different modalities must be available, all of which can be shown to contribute to the task.

---

[9] https://www.wikidata.org/wiki/Property:P21

[10] Even this is not a given. In many cases, the models themselves also amplify the biases present in the data[37].

This narrow range of requirements can lead difficult choices: in our search for suitable targets, we noted that the category "Sex or gender" in Wikidata, satisfied our technical requirements very well. However, training a model to predict this relation is to train (in part) a gender classifier, which is a controversial subject [11]. The following reasons have been posed for why such classifiers would be undesirable:

- Both sex and gender are not well captured by binary categories. Even the range of 36 categories offered by Wikidata (of which 8 are present in the Movie data) is unlikely to capture the spectrum of possibilities.
- People with gender identities outside the male/female categorization are at risk of oppression or discrimination. An oppressive regime may abuse gender and sex classifiers for large scale detention or prosecution. While there are currently no such systems employed to our knowledge, such practices do already exist in the related cases of race and ethnicity classification [18].
- The possibility of gender classification from external features may falsely imply a strong or causal relationship. Here, a comparable case is [33,14], where a classifier was built to predict sexual orientation. Besides the possibilities for abuse noted previously, such classifiers are often misinterpreted as showing strong causal links, for instance between physical features and the target class. In fact, all that can really be inferred is a weak correlation, which may well be based on incidental features, such as lighting, or personal choices such as clothing and make-up.

On the other hand, the inclusion of sex and gender as features in the data is important for the study of algorithmic bias. Simply removing the sex or gender attribute as a *target* class, but not as a feature of the data, also does not circumvent these issues. In a link prediction setting rather than a node labeling setting, every relation in the data becomes both feature and target. In such settings the two cannot be separated, and the problem remains.

Ultimately, we have chosen to include the dataset, with the "Sex or gender" attribute in place. We urge that practitioners use these datasets with care. For the gender-prediction task `mdgender` itself, we recommend strongly that this dataset be used only as a test case in development,[11] and not to report model performance in general settings, unless the task at hand is specifically relevant to the issue of sex or gender bias.

## 6    Conclusion

In this work, we have introduced a collection of multimodal datasets for the precise evaluation of node classification tasks on RDF-encoded knowledge graphs.

---

[11] The task in its current setup is too easy to serve as a good benchmark (which we have deliberately refrained from fixing). However, it is unique among these datasets in offering a strong guarantee that the images can be used to predict the target label with good accuracy. This property may be useful in debugging models, which can then be evaluated on the other tasks.

All datasets are available on GitHub and Zenodo in N-Triples and HDT format. Also provided are CSVs with an integer:label mapping, which can be loaded into Numpy and Pytorch by using the provided dataloader code. To support images, videos, and audio sequences, we also introduced a modest ontology to express these datatypes as binary-encoded string literals. For all datasets, we demonstrated their performance using several baseline models.

**Limitations** To add extra modalities to our data, we have relied primarily on images. Other modalities are available: for instance wikidata contains a rich collection of audio clips which provide an additional modality. Even small videos might be suitable.

An important consideration in constructing our graphs was to keep the total size of the graph relatively small. This means that the graphs presented here paint a slightly simplified image of real-world knowledge graphs. A model that performs well on these graphs can most likely not be applied directly to knowledge graphs found in the wild, as these will have magnitudes more relations, and relevant information stored more steps away from the instance nodes.

**Outlook** To stimulate adoption of the benchmark, we have aimed to offer a simple and unambiguous way to load the data (including baseline implementations for reference) and to host the data in multiple, redundant places (Zenodo and Github). As the data is used, we will offer a leader board on the Github page to track top performance and collect papers making use of the data.

The ultimate test of a benchmark task is whether it can be solved. In cases like speech-to-text, we can use human performance as an upper bound, but in a relational learning setting this is difficult to measure. Our baseline tests show that simple baselines reach low, but above-chance performance, with plenty of room for growth. It is difficult to establish what the performance ceiling is, but we hope that by providing a good number of datasets, we increase the probability that one of them will turn out to contain that particular trade-off between difficulty and simplicity that typifies the most enduring benchmark tasks.

Our ultimate hope is that these benchmarks stimulate more principled research towards models that learn end-to-end from relational and multimodal data, and that such models help to bridge the gap between statistical and symbolic forms of knowledge representation.

*Acknowledgements* We thank Emma Beauxis-Aussalet for illuminating discussions on the broader impact statement. We thank Rein van 't Veer for invaluable assistance with the geometry modalities in the DMG dataset.

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
