# OpenReview forum: "kgbench: A Collection of Knowledge Graph Datasets for Evaluating Relational and Multimodal Machine Learning"
_eswc-conferences.org/ESWC/2021/Conference/Resources_Track — ESWC 2021 Resources_

### Official Review · AnonReviewer1 · 2021-01-09
**Potentially interesting resource**

**Rating:** 2
**Confidence:** 4

**Review:**

This paper presents kgbench, a collection of datasets specifically designed for benchmarking of node classification in knowledge graphs. The main premise of the paper is that existing datasets are not good enough for this task since the size of their test sets tends to be not statistically significant. The paper makes special emphasis in message passing algorithms, like R-GCNs and R-GATs, which are particularly relevant for the node classification task, since such task cannot be solved directly using the node embeddings, unlike link prediction. On the contrary, message passing algorithms take into account the context of the node to learn the model, which makes them specially suitable for this task.

The datasets also over some multimodality aspect, focused on images. This creates opportunities in more diverse use cases, supporting two different evaluation scenarios: relational and multimodal setting. The authors also implement three baselines (feature-based, R-GCN and MR-GCN) on the provided datasets. I wish such baselines would have been evaluated also on the classical datasets used until now in node classification, e.g. CoRA, AIFB, AM or MUTAG.

This effort comes across as a good step in the right direction. The reengineering of the original datasets guaranteed well-balanced splits and class distribution. However, I miss a) more diversity in the domains covered by the datasets and b) datasets of some value in some economic sector. My understanding is that this would encourage further uptake (not only of these datasets but also of this family of machine learning models) beyond the academic community.

The discussion section is a little bit long-winded and looks overimposed to the rest of the paper. Additionally, a proper conclusion to the paper is missing.

**Anonymity:**

Yes, I would like my review to remain anonymous.

**Strong Points:**

- Potentially useful and necessary resource.
- The paper is well motivated and, in general, well written.
- The design of the datasets seems well informed and adequately balanced.
- Baselines are provided and the code and data is available on GitHub.

**Subreviewer:**

I submitted this review.

**Weak Points:**

- Not much diversity in the datasets. Interest for practitioners beyond academic circles may be limited.
- The discussion section is a little long-winded and proper conclusions are missing.
- I miss some words on how the authors plan to stimulate the adoption of the benchmark. The availability of the baselines is one step in that direction, though.

---

> ### Author Rebuttal · Authors · 2021-01-29
>
> We thank the review for their thorough review and their kind words. We hope the following answers all open questions.
>
> _The authors also implement three baselines (feature-based, R-GCN and MR-GCN) on the provided datasets. I wish such baselines would have been evaluated also on the classical datasets used until now in node classification, e.g. CoRA, AIFB, AM or MUTAG._
>
> The R-GCN baseline is the same as the originally published one (so the performance is the same as in the original paper). The MR-GCN does not provide any advantage over the R-GCN on these datasets as they are not multimodal (and using the literal information would change the task). The features baseline is similar to that used in the R-GCN paper, so we expect it has similar performance.
>
> _However, I miss a) more diversity in the domains covered by the datasets and b) datasets of some value in some economic sector. My understanding is that this would encourage further uptake (not only of these datasets but also of this family of machine learning models) beyond the academic community._
>
> While we certainly strove for a variety of domains, we must admit that we were only partially successful. This is partly due to the difficulty of finding existing open data that fit our constraints (see also [the rebuttal to review 3](https://openreview.net/forum?id=yeK_9wxRDbA&noteId=7Ng7usoQxQt)), and partly due to the domains available in the LOD cloud, where economic sectors are underrepresented.
>
> One way to achieve greater variety is to start with existing non-KG benchmarks (such as imagenet or the Netflix data), to transform it to a knowledge graph and to extend it with information from the LOD cloud. This is a challenging approach that is out of scope for the current paper, but certainly something we are interested in as future work.
>
> _The discussion section is a little long-winded and proper conclusions are missing._
>
> We will attempt to achieve a better balance in this respect in the final version of the paper, and add a stronger conclusion.
>
> _I miss some words on how the authors plan to stimulate the adoption of the benchmark. The availability of the baselines is one step in that direction, though._
>
> Our main strategies are the following:
>
> * Offering a simple and unambiguous way to load the data, in a single line of code, that is suitable for most uses.
> * Hosting the data both on Github and on Zenodo to ensure long-term availability.
> * Offering baseline implementations (as the reviewer noted), to demonstrate how the data should be used.
> * Keeping a leaderboard on the Github page to keep track of the current state of the art on this data.
>
> We will clarify this in the paper.

---

> > ### Comment · AnonReviewer1 · 2021-02-03
> > **Answer to rebuttal**
> >
> > Thank you for your comments. I will be looking forward to seeing how this dataset is adopted and evolves.

---

### Official Review · AnonReviewer3 · 2021-01-12
**A collection of data sets of multimodal graph data, including code for loading and baseline models.**

**Rating:** 1
**Confidence:** 4

**Review:**

Neural Networks which directly operate on the structure of a graph are a well-established research subject with promising results for interpretable machine learning. The two typical application for those Graph Neural Networks are link prediction and edge labeling. On the latter task models take the graph structure into account e.g. by using neighborhood aggregation algorithms to predict target labels for a set of unlabled nodes. To measure the performance of those algorithms, knowledge graphs with a large number of labeled entities in the test set are needed.
The authors propose a collection of datasets with the focus on measuring the accuracy of message passing algorithms on the task of node labeling. The semantic triples of the datasets can also be cut into big enough training- and test-sets for the task of link prediction, every dataset in the collection already has a canonical split. The proposed collection of datasets include knowledge graphs from different domains in CSV format and code to load the data into numpy and pytorch. Since the data is multimodal, complex datatypes are stored as Binary Large Objects using a small ontology providing datatypes to distinguish between the information types (images, video, audio). The design of the datasets seems well thought through and focused on broad usability especially on task of measuring accuracy. The data is easy to load and use, baseline experiments for different models are calculated and provided as usage examples in the code, and most of the datasets are constructed to fit into memory (12GB GPU, 64Gb CPU) training a baseline model. The datasets give the impression that a lot of work went into their construction: The most straightforward set consists of data from the wikidata movie domain combined with data from the internet movie database but more complex sets like the dutch monument graph consist of data combined from different sources wich was furthermore cleaned and corrected. Whereas most of the datasets consist of information concerning monuments, citations or collection items of a museum, the movie dataset contains the "sex and gender" property of Wikidata. The authors discuss the bias of the data and the implications behind this kind of attributes and relations as training targets.
The proposed resource is available, reusable and well documented. The paper has hardly any errors ("models know_n as", p. 1; "we extract(ion)", p. 11; "with a (the) Adam optimizer", p. 12) is well written and easy to follow. The design criteria for the datasets are conclusive and the data layout is clear.


**Anonymity:**

Yes, I would like my review to remain anonymous.

**Strong Points:**

•	The paper is technically sound, well written and organized
•	The presented collection of datasets fills the gap of available benchmark datasets for node labeling tasks with a big enough test set of nodes
•	The discussion of the bias in training data and the choice of target relations (e.g. gender) using representations concerning people leading to perhaps falsely implied causal relationships is a big plus


**Subreviewer:**

I delegated this review to a subreviewer.

**Weak Points:**

•	While CSV is an easy to use data format, storing larger information types like videos as binary large objects is perhaps not the best choice
•	The published dataset collection also includes the aifb, am1k, amfull and wd-people datasets, the first two can be used with the data loader. This should be mentioned in the paper or cleaned up accordingly in the code and the repository

---

> ### Author Rebuttal · Authors · 2021-01-29
>
> We thank the reviewer for their kind words and thorough review.
>
> _While CSV is an easy to use data format, storing larger information types like videos as binary large objects is perhaps not the best choice_
>
> We fully agree that the use of binary large objects has its limits. Our main consideration was to package the data in a manner that makes it most difficult to load the data incorrectly. We believe that a single RDF file with embedded binary data contributes to this purpose, but it may well be that if the binary data becomes too large, this solution is no longer feasible. If the approach of embedding large literals with RDF data finds wider adoption, we expect that new technical solutions will be developed to help this method scale to larger data.
>
> The problem is indeed compounded in the CSV format. We fully agree that our datasets are probably at or around the scaling limit for this solution, and if larger datasets are built, the CSV solution used here will likely no longer suffice.
>
> _The published dataset collection also includes the aifb, am1k, amfull and wd-people datasets, the first two can be used with the data loader. This should be mentioned in the paper or cleaned up accordingly in the code and the repository_
>
> We include aifb and am1k as reference datasets. These may be useful to test performance against earlier published results. We will mention them in the paper. The datasets amfull and wd-people will be removed.

---

> > ### Comment · AnonReviewer3 · 2021-01-31
> > **Thanks for the rebuttal**
> >
> > I read and acknowledge the rebuttal. No further questions.

---

### Official Review · AnonReviewer4 · 2021-01-13
**A valuable contribution to the field, lacking detailes**

**Rating:** 1
**Confidence:** 3

**Review:**

The paper proposes a valuable contribution: knowledge graph embedding field does suffer from having a limited set of benchmarks that are often of a poor quality (FB and WN). The paper will certainly have an impact on the community in that respect.

The datasets are diverse, coming from different domains, and interesting. It is exceptionally interesting that the knowledge graphs are multimodal and contain either textual or visual information. This has the potential to stimulate new research directions and challenge the existing methods.

The paper is generally easy to read and offers interesting insights.  I commend the authors for including a discussion about the broader impact; this is an important aspect of any resource paper. The authors have done a good job discussing the sensitive aspects of their work.

Though I generally see this work in a positive light, with the datasets being a strong contribution, I do find it lacking in certain aspects - discussion about the properties of the data and the choice of baselines.

*Properties of the data.

 The authors carefully explain their choices in constructing the data splits and show the basic statistics of each dataset. However, what I lack here, to make the paper an informative resource paper, is the discussion of qualitative and quantitative aspects of individual datasets. For instance, how do the datasets differ from each other (except for their domains)? Do they have certain properties that make them more suitable for a specific family of methods? How exactly were the datasets chosen? I believe such a discussion would strengthen the paper significantly. The paper offers an interesting discussion on the different information needed for node labelling and link prediction. Could you position the datasets in this respect?

One potential way to partially address these aspects would be to analyse more quantitative properties of datasets, such as density, degrees and similar. Various embedding methods are sensitive to such properties to a certain degree. Moreover, such information would offer better insight into how different the datasets differ.

*Baselines.

Fully understanding that this is a resource paper, I do believe that the baseline performance could have been established much better. The paper currently considers rather arbitrarily chosen baselines and does not elaborate on why those methods were chosen. The paper offers an explanation that they provide baselines to show how to use the benchmark. However, I believe the role of a resource paper is also to establish the baseline performance or, more precisely, inform the community how well the current methods perform on these datasets. This would clearly establish the benefit of introducing a novel benchmark. To do so, the paper should consider various commonly used methods, considering both node labelling and link prediction. Moreover, as the introduced benchmarks target both 'traditional' and multi-model knowledge graphs, it might also be beneficial to provide a baseline evaluation in both settings.


Finally, the authors should at least shortly discuss the relation with the following work with a similar goal:

Ye Liu, Hui Li, Alberto García-Durán, Mathias Niepert, Daniel Oñoro-Rubio, David S. Rosenblum. MMKG: Multi-Modal Knowledge Graphs. In Proceedings of the 16th Extended Semantic Web Conference (ESWC), Portoroz, Slovenia.

[After rebuttal]
I acknowledge that I have read the rebuttal. After reading the authors response and the other reviews, I still think the paper has holes that prevent it from being a very clear resource paper.




**Anonymity:**

Yes, I would like my review to remain anonymous.

**Strong Points:**

Interesting datasets that address a big problem in a community

**Subreviewer:**

I submitted this review.

**Weak Points:**

Lack of discussion on quantitative and qualitative aspects of the datasets
Weirdly chosen baselines

---

> ### Author Rebuttal · Authors · 2021-01-29
>
> We thank the reviewer for their thorough review and their kind words. We believe the following are the main questions and points of criticism to be addressed. All answers given will also be included in the final version of the paper:
>
> _However, what I lack here, to make the paper an informative resource paper, is the discussion of qualitative and quantitative aspects of individual datasets. For instance, how do the datasets differ from each other (except for their domains)? Do they have certain properties that make them more suitable for a specific family of methods? _
>
> The paper focuses primarily on the task of node classification. There is, so far, not a rich variety of models available for this task (partly due to a lack of suitable data sets). One property that we have tried to emphasize is that of a relatively small knowledge graph, so that the model may be trained with limited resources, which we believe benefits research.
>
> _How exactly were the datasets chosen? I believe such a discussion would strengthen the paper significantly. _
>
> We based our choices on the desiderata outlined in section 2.1. We then combed through the LOD cloud in search of datasets that could be modified to cover these desiderata, and that  would be likely to contain high quality data. Given the constraints that the graph should
>
> 1. be relatively small
> 2. contain a large number of instance nodes of the same type
> 3. have a meaningful and challenging labeling task on these instance nodes and
> 4. contain a rich variety of multimodal literals, whose content contains information about the labeling task,
>
> most existing knowledge graphs were unsuitable, and we were left with a small number of options.
>
> _The paper offers an interesting discussion on the different information needed for node labelling and link prediction. Could you position the datasets in this respect?_
>
> All datasets are suitable for both tasks, but our primary aim was to build datasets suitable for node labeling. This inspired many requirements, such as the high number of instance nodes and the need for a manageable size of graph (since node-labeling often requires full batch models).
>
> _One potential way to partially address these aspects would be to analyse more quantitative properties of datasets, such as density, degrees and similar. Various embedding methods are sensitive to such properties to a certain degree. Moreover, such information would offer better insight into how different the datasets differ._
>
> We agree that a set of such metrics would be useful to have. While embedding methods are not our primary focus, node labeling methods are likely also sensitive to these properties. We can include such a table in the github repository on publication.
>
> _Fully understanding that this is a resource paper, I do believe that the baseline performance could have been established much better. [...] it might also be beneficial to provide a baseline evaluation in both settings._
>
> We fully agree that our baseline experiments are of limited value. They serve primarily as a proof-of-concept of the methods involved. We would note that in other resource papers published at ESWC (for example [15]) such baseline experiments are entirely absent.
>
> We also note that a comprehensive baseline evaluation requires a thorough hyperparameter tuning for each model. To do this fairly, would require an automated hyperparameter search. In the case of link prediction, this suggests a program such as that followed in [1]. These experiments required several GPU-months to run, and were worthy of a publication in themselves.
>
> In short, a proper baseline evaluation, as valuable as it would be, is outside the scope of this paper.
>
> _Finally, the authors should at least shortly discuss the relation with the following work with a similar goal [...]_
>
> We will include such a discussion, and reference several other papers as well (see also [the reply to review 2](https://openreview.net/forum?id=yeK_9wxRDbA&noteId=lPSQ2F3_eQV)).
>
> [1] Ruffinelli, Daniel, Samuel Broscheit, and Rainer Gemulla. "You can teach an old dog new tricks! on training knowledge graph embeddings." International Conference on Learning Representations. 2019.

---

### Official Review · AnonReviewer2 · 2021-01-14
**A dataset paper that needs stronger ties to previous and related work to have impact**

**Confidence:** 5

**Review:**

This paper introduces benchmark datasets for evaluating node classification based on knowledge graphs.

The authors claim that research on relational and multimodal data is a challenge to assess because of the limited number of benchmarks and the way such benchmarks are built. Machine learning models such as graph neural networks are mentioned as promising solutions for solving multimodal problems, but the lack of datasets hinders further research progress. The reviewer is sympathetic to this. Further on, the authors introduce datasets as a challenging testbed and evaluate the performance of three baseline models on these datasets.

However, despite the interesting problem statement, it is not clearly understandable for the reader what the main contributions of the proposed tasks and datasets are with respect to the state of the art. Related work is missing in the paper and therefore the main problem statement (= lack of high-quality benchmark datasets) is not verifiable for the reader.

It is also important to mention that the paper repeatedly states assumptions and statements on different topics without substantiating them with relevant evidence from previous research or discussing how the authors reach a certain conclusion.
One example for such a statement, which can be found in the sub-section “Desiderata” is the following: “A good machine learning benchmark must satisfy a large number of constraints. We have focused primarily on the following.”

[After rebuttal] We acknowledge the rebuttal, but we feel that the arguments put forward in response to our most critical concerns (contribution, baselines, biases) have not been addressed in detail. The authors either aknowledge concerns only to dismiss them as out of scope, or promise to expand/improve the paper along those lines. The rebuttal is not specific enough and lacks detail to convince us that the paper would be substantially improved in the current review cycle.

**Anonymity:**

Yes, I would like my review to remain anonymous.

**Rating:**

-1: Weak Reject

**Strong Points:**

- identifying an area where more community work is needed to support research and advance beyond the state of the art.

**Subreviewer:**

I delegated this review to a subreviewer.

**Weak Points:**

In the following you will find weak points and suggestions for improvement:

The introduction misses related work on several topics which are addressed in the paper: graph neural networks, use of knowledge graphs in machine learning research, multimodal machine learning, knowledge graph/multimodal datasets introduced in the past.

It is hard for the reviewer to understand the contribution of the presented datasets as no other dataset is mentioned in the paper which uses multimodal data such as knowledge graphs, images or audio and could serve as a baseline. In recent NLP conferences e.g. EMNLP and ACL several new benchmarks and datasets for machine learning research based on knowledge graphs/multimodal data were introduced. How do those compare with what the authors set out to achieve.

The authors write about “message passing” but don’t explain the term as well as its links to their proposed tasks and datasets.

Some parts of the argument are one-sides. For instance in the desiderata the authors state: “Keeping the training set size relatively low has several benefits”. The rationales for a small training set size are listed, but without touching on the related disadvantages.

Which modalities to include in the datasets is an important decision and a valuable information for reviewers – this information is only stated at the end of Section 2 (“[…] we limit ourselves to three modalities which are popular in the machine learning community: images, videos, and audio sequences.”) and not mentioned anywhere else in the paper.  This decision needs to be communicated early and discussed more amply.

The description of the dataset creation process is very limited. No information about the annotation process, as well as the quality and evaluation of the annotated data is provided.  This limits the reproducibility and repeatability of the work and does not  give other researchers confidence in using the data and interpreting and contextualising the results of their algorithms on said data.

Baseline performance: The performance of the baseline models is shortly discussed in Section 4.1 but the authors miss comparing their results to any upper-bound measure e.g. human performance on the same task. Moreover, performance of the baseline models on other datasets is missing e.g. how the baseline model MR-GCN performs on other SOTA benchmarks. Such a comparison would allow the reviewer to understand the challenge of the proposed tasks in comparison to other relevant benchmarks introduced in past research.

In the last section of their paper, the authors briefly discuss the problem of bias in datasets. While bias in datasets is a common and important problem to be tackled, it is addressed very superficially in this paper. No previous research work on bias in datasets is mentioned.

In the last section the authors state “Our baseline tests show that our benchmark tasks are not easily solvable”, which is not inferable from the baseline results in our opinion.

Typo in first sentence of Section 3.3 “The Movie Data sets are a subset of Wikidata [21] in the movie domain.”

Typo in “In particular, the class of machine learning models know as message passing models offer an interesting set of abilities [23].”

Typo in the first paragraph of 3.4 “To provide a classification task on this data we extraction citation counts from the OpenCitations project [14], using the REST API.”

---

> ### Author Rebuttal · Authors · 2021-01-29
>
> We thank the reviewer for their thorough review. We have isolated the following as the most important points of criticism. We will amend the text to include our clarifications:
>
> _[...] the paper repeatedly states assumptions and statements on different topics without substantiating them with relevant evidence from previous research or discussing how the authors reach a certain conclusion._
>
> We will go through the paper to find and fix any unsubstantiated claims. In the specific case of the desiderata, these should not be read as claims: only as a subsection of the landscape of possible datasets that we have decided to focus on. We simply state our area of focus without any claim that other properties are less worthy of focus.
>
> _The introduction misses related work on several topics [...]_
>
> We will amend the introduction to give a more complete overview of the state of knowledge graph modeling by machine learning. However, we would note that this paper is purely a resource paper. As such a full overview of this area is outside the scope of the paper.
>
> _[Recently] several new benchmarks [...] were introduced. How do those compare with what the authors set out to achieve._
>
> We have indeed missed a number of published datasets that are somewhat related to our efforts. So far, we have found [1], which is a similar effort in the area of link prediction, but without the multimodal component. [2], which is a multimodal knowledge graph with no specific ML task attached and [3, 4] which provide some multimodal data for link prediction. We will include these and any other relevant work we find referenced therein in the final version of the paper.
>
> [1] Safaviet al arXiv:2009.07810 (2020).
> [2] Wang et al Big Data Research 22 (2020)
> [3] Pezeshkpouret al. arXiv:1809.01341 (2018).
> [4] Liu et al ESWC (2019).
>
> _The authors write about “message passing” [...]_
>
> The term message-passing network refers to neural networks that propagate information over the graph structure. This is in contrast to embedding models, which only model the information in single edges explicitly and can only infer wider graph structure implicitly. Because message passing networks explicitly "traverse" the graph to compute predictions, they are a promising model for using more of the rich structure of knowledge graphs (including multimodal literals).
>
> _Some parts of the argument are one-sides. For instance in the desiderata the authors state: “Keeping the training set size relatively low has several benefits”. The rationales for a small training set size are listed, but without touching on the related disadvantages._
>
> In this specific case we do not see any clear disadvantages. We will carefully check the paper for any case where we've made a one-sided argument, and provide a more balanced view.
>
> _Which modalities to include in the datasets is an important decision and a valuable information for reviewers [...]._
>
> In our choice of modalities, we have focused on thoseor which the machine learning community has provided the richest array of models to consume them, and those which were available in existing datasets.
>
> The section the reviewer points to here does _not_ list the modalities we included, but the modalities for which we introduce new datatype IRIs.
>
> _The description of the dataset creation process is very limited. No information about the annotation process, [...]._
>
> The production of these datasets involved no annotation. All were created by subsampling and joining existing datasets. All code is available to reproduce the data.
>
> _The performance of the baseline models is shortly discussed in Section 4.1 but the authors miss comparing their results to any upper-bound measure [...]._
>
> It is not clear how human performance can be measured on knowledge graph classification tasks. Note that instance nodes are connected to all information in the graph so human performance will likely depend a lot on the interface through which they view the graph. To correctly measure this is a complicated question outside the scope of this paper.
>
> There are no relevant benchmarks on which to test the MR-GCN model. The performance of the R-GCN model on other data (Table 1) is available in the original paper. The features baseline of that paper is similar to ours.  We are not aware of any node classification datasets for (multimodal) knowledge graphs that would provide relevant performance comparisons.
>
> _While bias in datasets is a common and important problem to be tackled, it is addressed very superficially in this paper._
>
> We agree that more work on dataset bias should be cited. Our aim was to discuss and substantiate our choice explicitly _not_ to tackle the existence of bias in the data, and to make the user aware of the existing biases.
>
> _“Our baseline tests show that our benchmark tasks are not easily solvable” [...] is not inferable from the baseline results in our opinion._
>
> We fully agree, and we will remove this sentence.

---

### Official Review · AnonReviewer5 · 2021-01-15
**A valuable collection of datasets for node classification and link prediction in RDF graphs.**

**Rating:** 1
**Confidence:** 3

**Review:**

The paper describes a resource called "kgbench" that is a collection of evaluation datasets primarily meant to be used for node labeling tasks on RDF graphs, although it can also be used for link prediction.

The datasets and the code to load them are all available on GitHub (https://github.com/pbloem/kgbench). The GitHub repository contains detailed documentation of the resource and how people can use it. Snapshots of the resource are published on Zenodo, and the corresponding DOIs are mentioned in the paper. It would be good if the DOIs of the Zenodo snapshots were clickable links in the paper, and/or included in the references.

The title of the paper sets the stage for a resource applicable to "Heterogeneous Knowledge Graphs", however only RDF graphs are considered, so I find that somewhat misleading. It would be great to augment the paper with a discussion of how/whether the resource could be used in property graphs.

The quality and clarity of the writing are excellent, and the contribution of the paper is unique and potentially very useful since there are no high-quality datasets fit for the same purpose. Overall I think kgbench is a valuable resource that is worthy of acceptance and presentation at ESWC.

**Anonymity:**

Yes, I would like my review to remain anonymous.

**Strong Points:**

- Excellent writing and structuring of the paper.
- Datasets are provided in RDF and CSV formats, along with Python libraries to load data in Pytorch and Numpy.
- Great documentation on GitHub, which includes explanations of how to load/use the datasets with the provided Python code.
- Good discussion of the biases in the datasets.

**Subreviewer:**

I submitted this review.

**Weak Points:**

- Lacking a discussion of the applicability of the resource to non-RDF knowledge graphs.

---

> ### Author Rebuttal · Authors · 2021-01-29
>
> We thank the reviewer for their kind words and thorough review.
>
> We expect that the use of the word heterogeneous (without a definition of terms) may have been confusing. By heterogeneous knowledge graphs we mean (following [23]) those with a variety of modalities encoded in their literals. However, in other contexts "heterogeneous graphs" is used to refer to a particular form of graph with typed nodes (and usually undirected edges). We will remove the phrase heterogeneous from the final version of the paper, or define it more precisely where we cannot remove it.
>
> We do indeed present knowledge graphs that fit the RDF framework only. We will clarify this in the text, and remove any false impression that we provide anything beyond that.
>
> For the specific case of using this data to test models for labeled property graphs, the data could be transformed. This task is however not entirely trivial (a solution would need to be found for the datatypes, for instance). We will include this in the discussion, together with notes on other types of graph structure that could be considered knowledge graph.
>
> We will make all URLs clickable, and include the Zenodo versions of the data as references.

---

### Decision · Program_Chairs · 2021-02-23

**Decision:**

Accept

**Comment:**

The paper introduces kgbench, set of new benchmarks for node classification on knowledge graphs. Overall, most comments are positive and the majority of reviewers is in favour of accepting the paper. However, Reviewer 2 advanced some concerns with regards to the state of the art and recent benchmarks. I believe that the paper makes a worthwhile contribution and that the issue can be fixed in the camera ready, as mentioned in the rebuttal. Therefore, I recommend accepting the paper. However, the authors will have to carefully address all comments of the reviewers and in particular improve the state of the art as suggested by Reviewer 2.